# Industrially Compatible Transfusable iPSC-Derived RBCs: Progress, Challenges and Prospective Solutions

**DOI:** 10.3390/ijms22189808

**Published:** 2021-09-10

**Authors:** Zhong Ri Lim, Svetlan Vassilev, Yew Wai Leong, Jing Wen Hang, Laurent Rénia, Benoit Malleret, Steve Kah-Weng Oh

**Affiliations:** 1Stem Cell Bioprocessing, Bioprocessing Technology Institute, Agency for Science, Technology and Research, Singapore 138668, Singapore; lim-zhongri@bti.a-star.edu.sg (Z.R.L.); svetlan_vassilev_from.tp@bti.a-star.edu.sg (S.V.); 2Agency for Science, Technology and Research Infectious Diseases Laboratories (A*STAR ID Labs), Immunos, Biopolis, Singapore 138648, Singapore; leong_yew_wai_from.tp@IDLabs.a-star.edu.sg (Y.W.L.); renia_laurent@idlabs.a-star.edu.sg (L.R.); 3Singapore Immunology Network, Agency for Science, Technology and Research (A*STAR), Immunos, Biopolis, Singapore 138648, Singapore; Benoit_Malleret@immunol.a-star.edu.sg; 4Department of Microbiology and Immunology, Immunology Translational Research Program, Yong Loo Lin School of Medicine, Immunology Program, Life Sciences Institute, National University of Singapore, Singapore 117545, Singapore; e0385049@u.nus.edu (J.W.H.);; 5Lee Kong Chian School of Medicine, Nanyang Technological University, Singapore 308232, Singapore

**Keywords:** erythropoiesis, human induced pluripotent stem cells, bioprocess intensification, reprogramming, enucleation, GMP-compliant, hematopoietic stem cells, terminal maturation, erythropoietic transcription factors, genetic amenability

## Abstract

Amidst the global shortfalls in blood supply, storage limitations of donor blood and the availability of potential blood substitutes for transfusion applications, society has pivoted towards in vitro generation of red blood cells (RBCs) as a means to solve these issues. Many conventional research studies over the past few decades have found success in differentiating hematopoietic stem and progenitor cells (HSPCs) from cord blood, adult bone marrow and peripheral blood sources. More recently, techniques that involve immortalization of erythroblast sources have also gained traction in tackling this problem. However, the RBCs generated from human induced pluripotent stem cells (hiPSCs) still remain as the most favorable solution due to many of its added advantages. In this review, we focus on the breakthroughs for high-density cultures of hiPSC-derived RBCs, and highlight the major challenges and prospective solutions throughout the whole process of erythropoiesis for hiPSC-derived RBCs. Furthermore, we elaborate on the recent advances and techniques used to achieve cost-effective, high-density cultures of GMP-compliant RBCs, and on their relevant novel applications after downstream processing and purification.

## 1. Introduction

Over the last century, society has relied heavily on blood transfusions for the restoration of blood oxygen capacity in various healthcare procedures, ranging from elective surgeries and chronic disorder treatments (such as thalassemia and several forms of anemia) to trauma surgeries. Therefore, a constant supply of donated blood of the different blood groups is essential for these lifesaving treatments. The main reasons as to why potential solutions for the global blood demand needed to be found are as follows: an overwhelming worldwide demand for blood, storage limitations of the donated blood units and a lack of potential blood substitutes.

### 1.1. Overwhelming Worldwide Demand for Blood

Although there are many blood banks globally with over one hundred million units of blood donated worldwide annually, the ever-changing world situation has resulted in consistent shortages. Factors that severely impact availability include peak holiday periods, natural disasters, military conflicts and unpredictable pandemics [1,2,3]. It was estimated that only 272 million units out of 305 million units of the global blood shortage in 2017 would be met, a shortfall of 33 million units [4]. While models could help predict the precedent need, pandemics are trickier to project. One example is how blood shortages have further intensified in the post COVID-19 world. In 2019, the global COVID-19 pandemic severely hit blood banks across the entire world, resulting in a 50% drop in blood units collect in Zambia, Singapore’s blood stocks declining by a third and 39% of the US’s blood centers having one- to two-day supplies [5,6], further exacerbating the pressing demand for blood. In light of this, alternatives to blood donations have to be seriously considered to cover for these supply shortages.

### 1.2. Storage-Induced Lesions in Donor Blood

Presently, the most commonly used protocol for red blood cells (RBCs) preservation is through removal of plasma, or sometimes even leukoreduction, into red cell concentrates to be stored in anticoagulant solutions (typically citrate–dextrose–phosphate). Thereafter, the product is stored in a slightly hypertonic additive solution, generally SAGM (sodium, adenine, glucose and mannitol, 376 mOsm/L) at a temperature of 4 ± 2 °C [7]. Despite the abundant units, the US Food and Drug Administration (FDA) mandates for RBCs to be kept for a period of no more than 42 days [8].

Generally, the reported mean age of the RBC units transfused is about 21 days [9], as prolonged storage would lead to altered RBC metabolism, lowered red cell viability, degradation of red cell membrane and higher oxidative stress [10]. This well-documented cascade of biochemical changes resultant from RBC refrigeration was found to alter the function of the RBCs detrimentally over time [11]. There was a positive correlation between the increased non-transferrin-bound iron (NTBI) following transfusion of older stocks of donated blood, which, in turn, resulted in higher infection risks and oxidative damage [12]. Hemolysis during prolonged blood storage was also found to be caused by decreasing concentrations of an allosteric effector, 2,3-diphosphoglycerate (2,3-DPG), and adenosine triphosphate (ATP), resulting in the loss of red cell membrane integrity and detrimental rheological property changes [13]. Additionally, there is a significant risk of developing hypocalcemia during massive transfusions of stored blood for trauma patients due to the chelation of the citrate and serum calcium (found in 97% of the transfused patients) [14].

However, these lesions and storage-induced alterations to the RBCs’ properties are partly reversible through methods such as supplementation of additive solution-7 [15] and rejuvenation solutions (which are adenine, inosine, phosphate and pyruvate based) to restore the intra-cellular ATP and 2,3-DPG levels [16]. Even so, these remedies come at extra time and cost, reducing their practicality.

### 1.3. Lack of Potential Blood Substitutes

Amidst the aforementioned challenges, there are groups of individuals (e.g., Jehovah’s Witnesses) that refuse blood transfusions, presenting a definitive need for alternative agents that could replace blood. Over the decades, perfluorocarbon (PFCs) emulsions and acellular hemoglobin-based oxygen carriers (HBOC) have been the two most developed substitutes among attempts to generate RBC replacements. Despite the amount of research done in this field, the half-lives of these molecules were found to be low in circulation and detrimental for health in vivo, due to vasoconstriction [17,18,19].

The first application of PFCs was shown through demonstrating the survival of mice immersed in an oxygenated PFC solution [20]. It is composed of inert carbon and fluorine atoms that have strong intramolecular bonds for stability, but weak intermolecular bonds for gas-like fluid properties and hence dissolve oxygen and carbon dioxide easily. Their practicality was enhanced by their ability to uptake and release O_2_ regardless of the environment and temperature. Fluosol-DA-20 was the only FDA PFC approved in 1989 but was subsequently withdrawn in 1994 due to low half-life of 12 h, low significance on survival and a magnitude of side effects [21]. Oxygent^TM^ and Oxycyte^TM^ were two other PFCs that were made but lacked approval from the FDA, with the former discontinued due to manufacturing costs and the associated increased stroke rates, and the latter terminated due to a lack of enrollment into phase II trials [22,23,24]. Despite not being FDA approved, Peftoran (rebranded Vidaphor in North America) was the only PFC that cleared clinical trials and has been approved for use in Russia so far. However, its usage remained low and only found significant potential for use in hemorrhagic shock in an event where human RBCs were unavailable [25].

On the other hand, cell-free hemoglobin is in theory a potential substitute due to its ability to transport O_2_ outside RBCs [26]. However, it was practically challenging to generate efficient HBOC due to the necessary-presence of 2,3-DPG for effective O_2_ transport and HBOC’s toxic and highly reactive nature. Owing to the adverse effects observed in clinical trials, the majority of the second generation HBOCs were terminated [27]. While none of the HBOCs were FDA approved, one (Hemopure) is available for use after exhaustion of all treatment methods for severe anemic patients through the expanded access program (compassionate use) by the FDA [28]. It was also approved by the South African drug council in 2001 for anemic treatments [29].

Despite several unsuccessful attempts in generating blood substitutes due to the aforementioned drawbacks, quality research efforts are still ongoing and show increasing promise for their application as blood substitutes [30,31,32]. Some notable studies for HBOCs that proceeded towards clinical trials are worth mentioning. In 2007, a novel HBOC with platelet-like activity, made by crosslinking hemoglobin with fibrinogen to polyhemoglobin-fibrinogen, showed favorable results in coagulation studies in vitro and in vivo and proceeded into phase III trials [33]. In 2018, a phase Ib, open-label, single arm study to assess another HBOC, SANGUINATE (SG), in patients with end-stage renal disease resulted in transient adverse events with no immune rejection throughout the study [34]. While several PFCs and HBOCs have found success in clinical trials, an emerging shift of focus towards stem cells for cultured RBCs took place.

## 2. Cell Sources for In Vitro Erythropoiesis

### 2.1. Hematopoietic Stem and Progenitor Cells (HSPCs)

Without potential blood substitutes to meet the demands required globally, early research into in vitro derivation of RBCs primarily focused on HSPCs when the clinical infrastructure for public banking of peripheral blood (PB) and cord blood (CB) began [35,36]. It was hypothesized that a single unit of CB can optimally produce more than 500 units of cultured RBCs (cRBCs) in the best case scenario [37,38,39,40,41,42].

Douay et al. first pioneered in vitro cRBC generation from CB HSPCs to about 200,000-fold increased pure erythroid populations (95–99%), with low enucleation (4%). However, they were able to achieve full in vivo terminal maturation in non-obese diabetic severe combined immunodeficient (NOD/SCID) mice with transfusion of these precursor cells [42]. Subsequently, they further improved the protocol with 10-fold erythroid cell amplification and 99% in vitro terminal maturation by co-culturing the cells with murine MS-5 stromal cells (mMS-5). Other groups began co-culturing their cells with other feeders, such as CB-mesenchymal stromal cells (MSCs) and bone marrow MSCs [43], and human macrophages generated from CD34+ cells differentiated from the same allogeneic donor [37]. Whilst the former group had a lesser fold increase, Fujimi et al. significantly improved the expansion of CB HSPCs to 3.52 × 10^6^-fold with the latter method.

Besides co-culturing methods, initial studies from Nakamura’s group developed an optimized stroma-free protocol that resulted in a 5.8 × 10^5^-fold expansion of the erythroid population with a 77.5% in vitro enucleation rate [44]. Subsequently, Timmins et al. were the first group to use a 2 L bioreactor to successfully expand CB CD34+ cells by 2.25 × 10^8^-fold over 33 days, with more than 90% enucleation [41]. In 2017, Zhang et al. were able to achieve a similar fold expansion of 2 × 10^8^-fold in a shorter 21 days by scaling up expansion from 25-T flasks to a bottle-turning device (HERAcell 240i) at a rotation of 0.75 U per minute [45].

Some other groups sought to improve the cell yields through the supplementation with novel compounds. In an unbiased screen in primary human HSCs, Boitano et al. identified a purine derivative, StemRegenin 1 (SR1), which led to a 50-fold increase in the CD34+ population, giving an erythroid cell expansion output of 1.23 × 10^6^-fold [46]. Baek et al. sought to optimize a stroma-free protocol of 95% enucleation by using poloxamer 188 (P188) as a hydrodynamic stress cytoprotective to maintain the cell membrane integrity during maturation [47]. Subsequently, melanocortins, such as ACTH39, ACTH24 and α-MSH, were identified to regulate enucleation through an MC5R-mediated signal, causing nuclear polarization and contractile ring formation [48].

Conversely, due to the lack of amplification potential, fewer studies were focused on PB. Douay’s group was the first to expand PB CD34+ cells with mMS-5 co-culture and found them to enucleate similarly to CB CD34+ cells despite a ten-fold lower amplification [38]. Upon transfusion of reticulocytes differentiated from their CD34+ PB cells into NOD/SCID mice, the cells matured morphologically towards cRBCs, with 68–81% of cells enucleating in vivo. [39]. In 2012, a large scale expansion of 10^4^-fold of PB CD34+ cells was conducted in agitated 2 L glass vessels at 15 rpm, obtaining 5 mL of packed reticulocytes [49]. They managed to optimize their protocol to achieve more than 10^5^-fold expansion of PB CD34+ cells and cRBCs were found to persist at a high percentage of 78 ± 9% 24 h into murine circulation while standard adult blood cells declined rapidly to only 49 ± 9% within the same timeframe of 24 h [50]. Recently, Heshusius et al. developed a defined good manufacturing practice (GMP)-grade protocol to expand pure erythroid cultures from PB mononuclear cells without the need for prior CD34+ isolation. They were able to obtain a 3 × 10^7^-fold increase in erythroblasts in 25 days with an enucleation rate of more than 90% in a 1L G-Rex bioreactor [51].

### 2.2. Immortalized Erythroblasts

Another strategy that certain research groups chose to achieve in vitro erythropoiesis was through the generation of immortalized adult erythroid progenitor cell lines. The advantages of this technique lie mainly in its potential to expand limitlessly with very lean culture conditions to generate the cRBCs desired. As compared to HPSCs, a great deal of resources and time can be saved from the need for repeated donations of CB and PB progenitors for expansion.

While most of the available immortalized erythroid lines were derived from patients with certain blood disorders; there is very little research that uses normal human erythroid lines for immortalization. In 2013, Hirose et al. transduced c-Myc and BCL-XL into erythroid progenitors using doxycycline (DOX)-inducible lentiviral vectors for immortalization [52]. While maturation was possible when the overexpressed genes were switched off, in vivo injection was needed for enucleation to happen. In the same year, a reliable protocol for immortalization of human erythroid progenitor cell lines through inducible Tet-On HPV16 E6/E7 lentiviral vectors was published [53]. The immortalized lines produced were found to be capable of terminal maturation with 25% in vitro enucleation. An immortalized line of CB-derived erythroblasts was also generated by ectopic expression of three genetic factors (c-Myc, Sox2, and a shRNA against the TP53 gene). While maintaining its undifferentiated state (i.e., without maturation towards reticulocytes/RBCs), it achieved a 10^68^-fold expansion over 12 months with 30% enucleation upon switching to maturation conditions [54]. In 2017, Trakarnsanga et al. successfully generated an immortalized line through inducible Tet-On HPV16 E6/E7 lentiviral vectors, which was able to yield up to 30% enucleation after 24 days of culture in a tertiary medium [55]. They went on to successfully generate a model disease immortalized erythroid cell line for hemoglobin E (HbE)/β-thalassemia, with about 10% enucleation [56]. Kurita et al. also immortalized several erythroid lines from bone marrow CD34+ cells through inducible Tet-On HPV16 E6/E7 lentiviral vectors [57]. Interestingly, the observed enucleation rates varied between cell lines obtained from the same initial CD34+ source, from 6.7% (BM-1-03) to 25.2% (BM1-01).

### 2.3. Human Embryonic Stem Cells and Human Induced Pluripotent Stem Cells

In the 2000s, human embryonic stem cells (hESCs) were a popular choice to study the process of human erythropoiesis. Early studies by several groups were focused on using a co-culture with various cell lines to derive hematopoietic cells (see Table 1). Murine bone marrow cell line S17 or yolk sac endothelial cell line C166 were first used on hESCs to demonstrate feasibility in 2001 [58]. Subsequently, murine stromal lines, such as OP9 and MS-5, were more commonly utilized for co-culture with hESCs, as they were distinctively efficient in the induction of hematopoietic differentiation [59,60,61,62]. In 2008, a landmark study demonstrated that hESCs were capable of differentiating towards the erythroid lineage, generating hemoglobinized erythroid cells with oxygen-carrying capacity [63]. Olivier et al. developed a procedure that co-cultured fetal human liver clone B cells (FH-B-hTERT) with hESCs to derive a more than 5000-fold increase in the erythroid population, which allowed enough cells for functional or biochemical experiments [64] (see Table 1).

However, the presence of feeder layers posed an additional problem for cell therapy-xenogeneicity. As such, there was a shift to stroma-free protocols for large-scale RBC generation as well-defined conditions are preferred. Lu et al. used a defined protocol comprising of an embryoid body (EB) culture, hemangioblast generation and expansion, followed by erythroid differentiation in a liquid medium to generate a more than 4000-fold expansion of RBCs with 10–30% enucleation [60]. Whilst the enucleation percentages were half those achieved when co-cultured with OP9, it was a good start towards stromal-free enucleation.

After the discovery of the Yamanaka factors (Oct3/4, Sox2, Klf4, c-Myc) and the development of human induced pluripotent cells (hiPSCs) through somatic cell reprogramming [65], a new era of erythropoiesis study was unlocked with healthy/pathological blood samples. In 2010, Lapillonne et al. pioneered the first differentiation of hiPSCs into erythrocytes using a suspension hEB protocol [66] (see Table 1). However, when compared to hESCs (30–65%), it had a very limited enucleating capacity (4–10%). They subsequently inhibited miR30A, which, in turn, increased the enucleation to 50%, with functional and mature RBCs expressing fetal and adult hemoglobin [67]. Kobari et al. was also able to prove recapitulation of ontogenetic hematopoietic differentiation by demonstrating that their hiPSCs were capable of terminal maturation and adult hemoglobin synthesis in vivo in NOD/SCID mice with 20–26% enucleation [68].

While small-scale suspension cultures were feasible for EBs, scaling up would prove more challenging. In 2016, Olivier et al. was able to optimize the EB differentiation process with the use of multiple novel small molecules to drive up erythroid expansion 5 × 10^4^–2 × 10^5^-fold from one O-negative hiPSCs in a T75 flask format with an enucleation efficiency of 10% [69]. In the same year, an integrated scale-up process by Wang et al. of multiple hiPSCs lines was also established in 1 L spinner flasks, with an enucleation efficiency up to 15% [70]. Initially, Sivalingam et al. managed to adapt several protocols from previous research through modulation of the Wnt/β-Catenin signaling and successfully expanded erythroid cells up to 10,000-fold with 28 to 40.6% enucleation with hMSC co-culturing [71]. However, certain stages of the protocol remain static and unscalable. Recently, Sivalingam et al. established a protocol for scalable differentiation for hiPSC-microcarrier aggregates in 500 mL spinner flasks, with about a 1000-fold expansion of the functional erythroblasts (see Table 1). Whilst hiPSC-derived erythroblasts exhibit low enucleation (6%), a co-culture with OP9 on the microcarriers resulted in up to 59.3% enucleation in vitro [72].

In the past couple of years, many research studies have tried to improve the enucleation percentages through supplementation of novel components and small molecules. Bernecker et al. attempted to replicate the physiological erythroid niche by forming a hematopoietic cell-forming complex (HCFC) and generated enucleated populations between 40 and 60% [73]. In another study, when KLF1-activated hiPSC-derived macrophages (iPSC-DMs) were found to increase the enucleated erythroid cell populations, Lopez-Yrigoyen et al. further investigated and found that three secreted factors (ANGPTL7, IL-33 and SERPINB2) were sufficient to replace feeders for 73% in vitro enucleation to occur in cord blood HSC-derived erythroblasts [74], and further work was to replicate this on hiPSCs. Olivier et al. further optimized their previous protocol for a chemically defined medium that was supplemented with FeIII-EDTA (see Table 1), an iron chelator to encourage transferrin recycling, ultimately improving the enucleation rates from 20 to 76.4% [75].

In the subsequent sections, we will focus on the challenges and their potential solutions for each specific stage involved in the generation of transfusable RBCs from hiPSCs.

**Table 1 ijms-22-09808-t001:** In vitro cRBC generation from hESCs and hiPSCs.

Reference	Cell Source and Cell Lines Used	Culture Platform (Vessel Format)	Feeder Cell for Enucleation	Novel Components for Erythroid Expansion or Enucleation	Culture Period	Fold Increase for Erythroid Diff’n	Percentage of Enucleated Cells (In Vitro)
Kaufman et al., 2001 [58]	hESCs (H1,H1.1 and H9.2)	Monolayer	S17, C166	-	18	Not reported	Not reported
Vodyanik et al., 2005 [61]	hESCs (H1 and H9)	Monolayer	OP9, S17, mMS-5	-	9	Not reported	Not reported
Olivier et al., 2006 [64]	hESCs (H1)	Suspension	FH-B-hTERT, mMS-5	-	39	5000 fold	No enucleation
Lu et al., 2008 [60]	hESCs (MA01)	Suspension	hMSCs, OP9	-	42	>10^4^-fold	10–30% feeder-free, 30–65% with OP9
Ma et al., 2008 [63]	hESCs (H1)	Monolayer	mFLSCs	-	18	100-fold	11%
Klimchenko et al., 2009 [59]	hESCs (H1 and H9)	Suspension	OP9	-	14	Not reported	Not reported
Lapillonne et al., 2010 [66]	hESCs (H1), hiPSCs (IMR90 and FD136)	Suspension	-	5–10% Human Plasma	26	~3500-fold, ~225- to 440-fold	52–66% 4–10%
Dias et al., 2011 [62]	hESCs (H1), Transgene/free hiPSCs (SK46-M4-10,Foreskin-1, 19-9-7T and 4-3-7T)	Monolayer	OP9, mMS-5	-	70–90	~4000-fold	2–10%
Kobari et al., 2012 [68]	hiPSCs (PB04 cell line from SCD)	Suspension	-	-	52	-	20–26%
Rouzbeh et al., 2015 [67]	hESCs (H1 and H9)	Suspension	-	-	34	75-fold	~50%
Olivier et al., 2016 [69]	hiPSCs (33D6)	T75 Flask	-	CHIR99021, Activin A, IBMX, SR1 and Pluripotin	31	2 × 10^5^-fold	10%
Wang et al., 2016 [70]	hiPSCs (BC1, TNC1 and E2)	1L S.Flask	-	-	29	240- to 370-fold	2–15%
Sivalingam et al., 2018 [71]	hiPSCs (IMR90, BR2, BR7, D5, D9, D11, D12 and X13)	Suspension	hMSCs	CHIR99021	42	>10^4^-fold	28–40.6 %
Bernecker et al., 2019 [73]	hiPSCs (CD34-iPSCs and PEB-iPSCs)	Monolayer	HCFC	-	56	100- to 1000-fold	40–60%)
Lopez-Yrigoyen et al., 2019 [74]	hiPSCs (SFCi55 and SFCi55-iKLF1.2)	Suspension	KLF1-activated iPSC-DMs	ANGPTL7, IL33 and SERPINB2	28	-	73% (CB HSC-derived erythroblasts) 6% (iPSC-derived erythroblasts)
Olivier et al., 2019 [75]	hiPSCs (NY22, OM1, OM2, OM3. and OM4)	Suspension	-	RED + and FeIII-EDTA	39	~1000-fold	20–76.4%
Sivalingam et al., 2021 [72]	hiPSCs (IMR90, BM1, CB6, FR202, BR7, D9, D12 and X13)	500 mL S.Flask	OP9	SR1 and Pluripotin	35	~1000-fold	6% *w*/*o* OP9, 18.1–59.3% with OP9

Abbreviations: ANGPTL7, Angiopoietin-related protein 7; BM, bone marrow; C166, embryonic day 12 mouse yolk sac; CHIR99021, GSK3b inhibitor; Cmpts, Components; Diff’n; Differentiation; FeIII-EDTA, ferric EDTA; FH-B-hTERT, fetal human liver clone B human telomerase catalytic subunit gene-transduced stromal cells; HCFC, hematopoietic cell-forming complex; hESCs, human embryonic stem cells; hiPSCs, human induced pluripotent stem cells; hMSC, human mesenchymal stromal cell; IBMX, isobutylmethyl xanthine; IL33, interleukin33; mFLSCs, mouse fetal liver stromal cells; mMS-5, murine marrow stromal cell line MS-5; MNC, mononuclear cells; MSC, mesenchymal stromal cell; P188, pluronic F68; S17, mouse bone marrow; SCD, sickle cell disease; S.Flask, spinner flask; SERPINB2, Serpin Family B Member 2; SR1, stem regenin 1; w/o, without.

## 3. Approaches to Enhance Erythroid Differentiation from hiPSCs

hiPSCs are a remarkable resource and have the potential to overcome many challenges for in vitro cell production, not just in the field of erythroid cells. Since the publishing of Yamanaka’s reprogramming work 15 years ago [65], publications detailing the techniques and tools for hiPSC generation, handling and differentiation have skyrocketed, with new developments being introduced in the field every year. We review the extensive technical challenges to successful commercialization of hiPSC-derived RBC production.

### 3.1. Reprogramming to hiPSCs

Cellular reprogramming to a pluripotent state rests on the observation that a few key transcription factors (TFs) expressed in an ectopic manner in somatic cells are capable of reverting their cellular identity to a state of pluripotency. Takahashi and Yamanaka’s work provided the first insights into this process using the now famous OSKM factor combination—Oct4, Sox2, Klf4 and c-Myc [65]. Nowadays, these are the most commonly used factors in reprogramming, although other combinations have also been documented; for instance, Thomson devised a reprogramming protocol with a slightly altered TF expression, using Oct4 and Sox2, but also Lin28 and Nanog [76,77].

Studies are still ongoing to determine the exact mechanisms by which somatic cells abandon their mature fate and revert to a pluripotent state. It is now known that expression of the aforementioned factors results in vast changes in chromatin structure and an altered transcriptional landscape. A more in-depth analysis performed by Stadtfeld and Hochedlinger in 2010 suggested the emergence of transcriptional waves in response to reprogramming. They described that c-Myc may act indirectly by priming cells and facilitating Sox2 and Oct4 binding, which initiate expression of endogenous pluripotency factors. Thereafter, chromatin reorganization and epigenetic modifications lead to the silencing of ‘mature’ genes [78]. Consistent with this notion, Markoulaki and colleagues demonstrated an increased fibroblast reprogramming efficiency when c-Myc expression was induced early before the remaining factors [79].

Reprogramming methods have advanced concurrently with our knowledge of the transcription factors involved. Early strategies used viral integrating methods for TF expression, which is unsuitable for a clinical therapeutic due to the risk of random mutagenesis. Accordingly, modern techniques have been designed to be much safer and more efficient. Here, we provide a brief overview of the reprogramming methods and their main strengths and drawbacks.

#### 3.1.1. First Generation: Integrating Methods

The earliest somatic cell reprogramming technologies relied primarily on integrating viral vectors for TF delivery and expression. Retroviruses in particular have seen extensive use due their well-understood biology, usually as replication-defective cassettes to maximize space for the desired TF payload [80]. However, it was soon discovered that most mammalian cells possess the capacity to silence retroviral gene expression through methylation of DNA [81], leading to a transient, short-lived expression of the reprogramming factors. Some groups have also shown compromised reprogramming and limited repression of somatic genes with these vectors [82]. Another common option for integrative viral delivery is using lentiviral vectors, which are capable of targeting more cell types and can transduce both non-dividing and dividing cells (as opposed to retroviruses, which can only infect dividing cells) [78]. Lentiviral reprogramming can achieve efficiencies of ~1% and has been modified extensively, for example, using polycistronic expression cassettes equipped with self-cleavage sequences [83]. Such systems can carry all reprogramming TFs on a single vector, thus simplifying reprogramming, increasing efficiency, and lowering the risk of silencing [84].

Finally, researchers have also attempted to develop reprogramming methods that do not rely on viral delivery systems, citing safety reasons and easier clinical development pathway [85]. Towards this goal, nonviral integrative solutions, such as mobile genetic elements (e.g., PiggyBac transposon technology and the Sleeping Beauty technique), have emerged as possible alternatives. These have been described extensively elsewhere [86,87,88,89].

#### 3.1.2. Second Generation: Non-Integrating Methods

Integration of genes into the host cell’s genome inherently carries with it a risk of gene disruption, as the site of integration is difficult to control. Integration can potentially disrupt essential genes, leading to undesirable mutations or oncogene activation. This has prompted researchers to develop non-integrative methods of expressing the desired reprogramming factors [90]. The first viral vector delivery of OSKM factors without integration has been demonstrated using adenovirus [91], which shows broad cell tropism. However, this approach is plagued by a low reported reprogramming efficiency, ranging from 0.0006 to 0.001% [83,92]. Additionally, studies have shown that adenoviral vectors are swiftly eliminated during cell division, resulting in low TF expression [80].

Other viral vectors, such as the non-integrating Sendai virus system (SeV), could overcome these challenges. The SeV approach has been developed into kits that allow researchers to transduce cells with replication-capable RNA molecules containing the desired reprogramming factors in a streamlined, easily replicable fashion. First proposed by Fusaki et al. in 2009, this technology has shown great promise due to its high relative efficiency and ease of implementation [93,94]. Other approaches have also seen use, including episomal self-replicating vectors [76] and direct mRNA transfection. Advantages of both include a better hiPSC consistency, ease of use and potentially easier clinical translation [95], although the latter does have shortcomings in terms of its reprogramming efficiency [96]. Notably, several studies have demonstrated that reprogramming efficiency can be boosted using additional factors coupled with OSKM expression, such as BCL-XL [97].

#### 3.1.3. The Use of Small Molecules

As pluripotent stem cell therapies have taken their first steps towards clinical application and commercial application, the cost of reprogramming has become an important consideration in the eyes of many groups. Current generation techniques and kits are relatively costly and produce hiPSCs at a fairly low efficiency. Thus, studies have been conducted aiming to enhance these techniques with cheaper small-molecule chemicals. For instance, Hou and colleagues performed a landmark study in 2013 in which they demonstrated reprogramming of murine somatic cells using a combination of seven small molecules, achieving an efficiency of up to 0.2% [98]. Building up on this work, in 2015 the same group boosted the efficiency up to 1000-fold by the addition of a further four compounds [99].

#### 3.1.4. Concerns Regarding Cell Quality

A common question in hiPSC research is whether somatic cell reprogramming manages to fully reverse the cell fate towards a “true” embryonic state, or simply mimics key features while failing to induce true pluripotency. Studies over the last decade have been inconclusive, but newer and more sophisticated research has provided strong support that hESCs and hiPSCs share almost all common features. For instance, Stadtfeld et al. performed an assessment of global RNA expression in murine ESCs and iPSCs and found a virtually indistinguishable gene expression landscape (with the exception of several transcripts on chromosome 12qF1) [100]. Guenther et al. also failed to detect any significant differences in patterns of histone modifications between the two pluripotent cell types [101].

Another key issue under intense investigation is also whether hiPSCs retain some form of epigenetic memory of their tissue of origin. Kim and colleagues analyzed the differentiation ability of murine fibroblast- and bone marrow-derived iPSCs and reported that BM-iPSCs showed higher expression of blood lineage genes and formed hematopoietic colonies more easily [102]. The same group followed up this study with an assessment of the hematopoietic vs. keratinocyte potential of hiPSCs derived from umbilical cord blood and mature keratinocytes, showing once again that UCB-hiPSCs had 9 times lower expression of key keratinocyte genes compared to the keratinocyte hiPSCs [103]. However, this should not be taken necessarily as a negative feature—on the contrary, this may aid researchers to produce high-quality cells of various lineages by reprogramming cells from the intended target tissue (e.g., reprogramming cord blood to make hematopoietic cells).

### 3.2. Generation of Hematopoietic Cells In Vitro

#### 3.2.1. Directed Differentiation with Growth Factors and Small Molecules

hiPSC-derived blood therapeutics have experienced strong development over the last decade, with many groups reporting the successful differentiation of pluripotent cells towards the hematopoietic and erythroid lineages [66,104,105]. Current generation protocols starting from hiPSCs generally follow a similar differentiation path that broadly recapitulates embryonic development [106]. Thus, we can define several key stages of commitment towards the desired hematopoietic (and eventually erythroid) lineage: mesoderm patterning, hematopoietic progenitor induction, erythroid commitment and terminal maturation [69,107].

Human hematopoiesis in the embryo begins with the formation of the primitive streak (PS) following the epithelial-to-mesenchymal transition of epiblasts from which the endoderm and mesoderm are specified [108], and is commonly associated with the expression of T-Brachyury (T-Bra), a T-box transcription factor [109]. Tam and Loebel demonstrated that various regions of the PS result in the induction of different lineages, with the most posterior section of PS leading to the development of hematopoietic mesoderm progenitors and, eventually, hematopoietic cells [110]. The bone morphogenic protein (BMP), Wnt and Activin/Nodal protein families have been implicated as key regulators of PS and mesoderm induction [111,112]. A recent study by Shen et al. demonstrated that while PS and hematopoietic mesoderm formation is mostly dependent on Activin and Wnt pathway stimulation, lack of BMP4 signaling abrogates hematopoietic differentiation [113]. Mesoderm cells committed towards the hematopoietic lineage are typically identified by cell surface receptors such as KDR and APLNR [114].

KDR+ progenitors are believed to eventually lead to the development of endothelial progenitor cells, which have the capacity to undergo so-called endothelial-to-hematopoietic transition (EHT) [115,116]. Hematopoietic progenitors generated by this process are identified by coexpression of CD34 and CD43 and can give rise to a variety of myeloid cells [117]. Differentiation along these pathways towards the erythroid lineage is tightly controlled via a complex network of cytokines and signaling molecules, including SCF, IL-3 and 6, TPO, FLT3L, IGF2 and many others [69,107] (see Table 1). Co-culture with feeder cells has also been shown to improve the hematopoietic output of hiPSCs [118]. Both OP9 and C3H10T1/2 feeders have seen use as supportive cells for differentiation [119,120,121]. Finally, the use of small molecules to enhance HSPC generation could be a potential solution to expensive cytokine cocktails. Compounds such as SR1, UM171 and UM729 have been used extensively in differentiation studies to promote hematopoiesis [122,123,124].

#### 3.2.2. Recapitulation of Primitive vs. Definitive Hematopoiesis

Although many current approaches are capable of generating HPSCs that give rise to hemoglobinized and enucleating erythroid cells, whether such processes proceed along primitive or definitive programs is yet to be firmly established. In mammals, these comprise the two distinct “waves” of hematopoietic development. The primitive wave, which serves to provide blood oxygenation in the developing embryo, is first observed in the yolk sac in so-called blood islands and its cells are transient in nature. As the embryo develops, definitive HSCs arise in the embryonic aorta–gonad–mesonephros (AGM) region, migrating afterwards to the fetal liver and finally establishing the hematopoietic niches in the bone marrow [125]. Directed differentiation of hiPSCs using cytokines and small molecules results in HSCs that generally lack engraftment potential [126], where engraftment is considered as the ability of HSCs to home in to bone marrow niches and survive and proliferate there for an extended period of time, thus giving rise to all downstream blood lineages following transplantation [127]. To this end, Suzuki et al. produced engraftable HSCs by way of in vivo teratoma formation [128]. However, this approach has clear shortcomings in its adaptability towards the clinical setting. The activation of specific transcription factors via genetic manipulation or small molecules could help alleviate this issue. Recently, Sugimura and colleagues demonstrated that seven factors (*SPI1, HOXA5, HOXA9, HOXA10, RUNX1, LCOR* and *ERG*) are capable of producing HSCs that are engraftable in both primary and secondary recipients [129]. Additionally, while many groups have demonstrated effective generation of functional erythrocytes from hiPSC-derived HPSCs, they have reported high levels of fetal (HbF) and embryonic (HbE) globins and low levels of adult globins, along with low levels of terminal enucleation [60,130]. Thus, efforts to understand the precise molecular pathways that govern primitive versus definitive hematopoiesis are ongoing. A study by Kennedy et al. in 2012 suggested that definitive hematopoiesis is promoted via timely activation of the Wnt pathway and inhibition of Activin/Nodal signaling [131]. Sturgeon et al. corroborated these findings and defined a KDR+CD235a- population of definitive progenitors generated in the early stage of mesoderm patterning [132]. These studies, as well as others investigating definitive differentiation, relied on T-cell potential as a marker of definitive hematopoiesis, as it is known that early primitive hematopoietic waves lack this capacity [133].

#### 3.2.3. Transcription Factor-Mediated Conversion to HSCs

Some groups have theorized that a more efficient system for hematopoietic cell generation could ideally bypass the lengthy hiPSC stages and mesoderm differentiation. Instead, using ectopic TF expression in adult fibroblasts or CD34+ cells could in theory result in direct reprogramming of somatic cells into HSCs and erythroid progenitors. Using 4 TFs (FOSB, GFI1, RUNX1 and SPI1), Sandler et al. was able to convert endothelial cells to HSCs with engraftment potential, with both myeloid and lymphoid capacity in vivo [134]. However, endothelial cell co-culture of the reprogrammed erythroid progenitors was found to be necessary, restricting the scalability and clinical potential of this method. In another study, GATA1, Tal1, LMO2 and c-Myc were sufficient to transform human fibroblasts into erythroid cells [135]. However, these factors alone were only capable of inducing embryonic globin expression, with additional overexpression of TFs needed to induce adult globin switching. Consequently, more efforts are needed to fully determine the optimal genetic perturbations for direct conversion methods to hematopoietic stem cells and erythroid cells.

### 3.3. Erythropoiesis

During definitive erythropoiesis, the earliest committed erythroid progenitors are the burst-forming unit erythroid (BFU-E) cells, which differentiate into colony-forming unit erythroid (CFU-E) cells. BFU-E and CFU-E cells are morphologically indistinguishable from other blast cells of non-erythroid lineages, but are identifiable based on their surface marker expression [136]. CFU-E cells further differentiate into proerythroblasts, the earliest morphologically distinct erythroid stage. Proerythroblasts mark the start of terminal erythroid differentiation, which begins with a series of mitoses (four in humans [137] and three in mice [138]) to form basophilic, polychromatic and orthochromatic erythroblasts (see Figure 1). Erythroblast differentiation is characterized by a decrease in cell size, nuclear condensation and hemoglobin accumulation. Young erythrocytes, or reticulocytes, are formed when orthochromatic erythroblasts undergo enucleation. Reticulocytes then migrate into the blood circulation to complete their maturation into biconcave normocytes. The major bottlenecks in recapitulating in vivo erythropoiesis in an hiPSC system are low erythroid progenitor expansion and differentiation, and a poor enucleation rate. Addressing these problems is key to the large-scale generation of iPSC-derived erythrocytes for clinical transfusion, and various solutions have been proposed (see Figure 1).

#### 3.3.1. Co-Culture with Feeder Cells—Mimicking the Bone Marrow Microenvironment

Besides enhancing hematopoietic differentiation of hiPSCs, co-culture with feeder cells has also been shown to improve terminal erythroid differentiation of iPSC-derived erythroid progenitors. Human iPSC-erythroblasts expanded more robustly when co-cultured with mouse MS-5 bone marrow stromal cell line in a cytokine-free medium, compared to iPSC-erythroid cells cultured in a feeder-free low-adherence environment in the presence of erythropoietin (EPO) and stem cell factor (SCF) [62]. Additionally, the co-culture condition yielded a higher frequency of late-stage erythroblasts and even some levels of enucleation (see Table 1). Another mouse bone marrow stromal cell line, OP9, was able to reduce apoptosis of expanding iPSC-erythroblasts and improved enucleation by about three- to ten-fold [72]. More importantly, the authors also showed that OP9 can be grown on microcarriers (microbeads that greatly improve the growth area of adherent cells in a suspension culture) and thus demonstrating the scalability of their iPSC-erythrocyte platform.

Shen et al. also observed an improved enucleation rate of human embryonic stem cell (ESC)-derived erythroblasts when co-cultured with OP9, but the enucleation rate remained low (around 8%) [121]. They noted that hematopoietic stem cells (HSC) arise from hemogenic endothelial cells in an endothelial microenvironment during embryonic development [139], so they included an endothelial cell (EC) co-culture step early in their differentiation protocol, prior to co-culturing the EC-primed erythroblasts with OP9. This sequential co-culture system (with EC then OP9) generated more ESC-derived enucleated erythrocytes when compared to a co-culture with EC or OP9 alone, or the reversed sequence (OP9 then EC). They verified the importance of this co-culture sequence by replicating the efficient enucleation rates (around 60%) with iPSC-derived erythroblasts. It is thought that a co-culture with ECs primed the erythroid progenitors with high enucleation potential, and the downstream OP9 co-culture further promoted differentiation and enucleation of the EC-primed erythroblasts.

#### 3.3.2. Co-Culture with Macrophages—Mimicking the Erythroblastic Island

Erythroblast expansion and maturation takes place in association with bone marrow macrophages, forming structures termed erythroblastic islands (EBI), comprising 5 to more than 30 erythroblasts surrounding a central macrophage [140]. Central macrophages support the development of erythroblasts through a variety of adhesion molecules mediating cell-to-cell contact and secreted factors that promote differentiation and enucleation of erythroblasts (reviewed in de Back et al. [141]). Other functions of the central macrophages include acting as a ferritin iron source for hemoglobin synthesis [142] and phosphatidylserine-dependent phagocytosis of the extruded nuclei [143].

Attempts to reconstruct EBIs in vitro have been demonstrated using primary erythroid progenitors co-cultured with macrophages derived from bone marrow and cord blood CD34+ HSCs [144], and from CD14+ monocytes in peripheral blood mononuclear cell (PBMC) fractions [145,146]. However, the role of macrophages in these in vitro EBIs is unclear. Macrophages derived from CD34+ HSCs were able to form EBIs with primary erythroblasts in vitro, and improved the erythroblast proliferation [144]. Heideveld et al. also observed increased erythroid yield from CD34+ HSCs when co-cultured with monocyte-derived macrophages [145]. Further investigation, however, revealed that the macrophages did not directly affect erythroblast expansion, but instead improved CD34+ HSCs survival and proliferation, resulting in an indirect increase in erythroid yield. In contrast, another study found that macrophages derived from CD34- PBMCs directly influenced adult HSC-derived erythroblasts [147]. When the erythroblasts were co-cultured with macrophages, fewer apoptotic erythroblasts were observed, and the overall erythroblast proliferation improved. Interestingly, co-cultured erythroblasts had lower enucleation rates, indicating that macrophages promote erythroid proliferation but have an inhibitory effect on erythroblast differentiation in diseases such as beta thalassemia.

These contradictory observations might be due to the different sources of the macrophages and the heterogeneity of the macrophage population that were used. It has become increasingly clear that tissue-resident macrophages, such as EBI macrophages, have a distinct developmental ontogeny compared to monocyte-derived macrophages [148], and this might severely limit monocyte-derived macrophages to reconstruct functional EBIs. iPSC-derived macrophages have been shown to closely resemble tissue-resident macrophages [149]; therefore, iPSC-macrophages might be a better candidate to reconstruct EBIs in vitro. Lopez-Yrigoyen et al. were successful in generating iPSC-macrophages that mimic EBI macrophages (based on their surface marker expression and phagocytic activity) via inducible activation of KLF1 [74] (see Table 1). These KLF1 iPSC-macrophages formed close associations with cord blood-derived erythroblasts and resulted in a higher yield of enucleated erythrocytes when compared to control iPSC-macrophages. This increase in yield was a consequence of better cell proliferation and viability, and also improved terminal differentiation. Furthermore, they replicated these results with iPSC-derived erythroblasts (instead of cord blood-derived erythroblasts) and therefore were able to completely reconstruct the human EBI in vitro using only hiPSC sources.

Although most of the abovementioned EBI studies did not use iPSC-derived erythroblasts (except for Lopez-Yrigoyen et al.), the preliminary insights gained from them are vital to informing future attempts at recreating a fully iPSC-derived EBI in vitro. Arguably, macrophages and the formation of EBI might play a bigger role in iPSC-erythroid development compared to adult HSC-derived erythroblasts. In vitro culture of adult and CB HSC-derived erythroblasts were able to result in exceptional enucleation rates (>90%) without macrophage support [38], and even in the absence of feeder cells (>70% enucleation rate) [44]. Contrastingly, mouse primitive erythroblasts failed to enucleate unless co-cultured with macrophages [150]. In humans, primitive erythrocytes are closely associated with placental macrophages during the first trimester, and enucleation is thought to be promoted by placental macrophages [151]. It is possible that macrophages are essential for enucleation during primitive erythropoiesis and considering that iPSC-erythroblasts display primitive-like characteristics, it is worth further investigating the role of macrophages in iPSC-erythroid proliferation, differentiation and enucleation.

#### 3.3.3. Genetic Approaches to Expand Erythroid Cells

Besides general approaches such as feeder cell and macrophage co-culture, a more precise approach to improve iPSC-erythroid generation is to compare the transcriptomic and proteomic profile of the iPSC-erythroblasts with erythroid cells differentiated from other sources. Identifying these differences and understanding how they impact erythropoiesis are key to employing genetic approaches to ‘correct’ for these aberrations, resulting in iPSC-erythroblasts that resemble functional adult erythrocytes.

The proliferative capability of iPSC-derived erythroid progenitors is found to be significantly lower than primary erythroblast progenitors derived from adult blood and cord blood [152]. Comparing the transcriptomes of iPSC-derived and adult erythroid cells, Merryweather et al. noted the relative downregulation of c-Kit in iPSC-erythroblasts. The low expression of c-Kit, which is the receptor SCF, is thought to result in the failure of iPSC-erythroblasts to expand in response to SCF in culture. Based on this premise, they induced ectopic expression of c-Kit in iPSC-erythroblasts via a lentiviral expression system and managed to slightly increase their proliferation. This moderate increase in proliferation, however, was still significantly lower than the proliferation of adult erythroid cells and could be due to the still relatively low c-Kit expression despite the lentiviral construct. Therefore, more efficient methods to express functional c-Kit on the surface of iPSC-erythroblasts could potentially overcome the proliferation bottleneck.

Another approach is to target the transcription factors that regulate erythropoiesis. In the same study, the expression of one of the key erythroid transcription factors, c-Myb, was found to be drastically reduced in iPSC-erythroblasts compared to adult erythroid cells [152]. Considering that c-Myb is essential for c-Kit expression in mouse erythroid progenitors [153], ensuring c-Myb expression in iPSC-derived cells could be a better alternative. Besides c-Kit expression, c-Myb also promotes erythroid lineage commitment by activating the erythroid transcription factors KLF1 and LMO2 [154]. Silencing c-Myb in human CD34+ HSCs resulted in increased megakaryocyte commitment and concomitantly impaired erythropoiesis; this impairment was partially rescued by constitutive expression of either KLF1 or LMO2. The same group also showed that c-Myb regulates erythroid lineage commitment by downregulating MAF via the microRNA miR-486-3p [155]. Collectively, these studies support c-Myb as a key erythroid regulator and mediates the expression of many downstream genes essential for erythroid commitment and differentiation. Ensuring c-Myb expression during erythroid differentiation of hiPSCs could be a promising strategy considering that c-Myb is poorly expressed by iPSC-erythroblasts [152].

Downstream of c-Myb, KLF1 is an important erythroid transcription factor, but it was found to be downregulated in erythroid cells derived from hESCs compared to those derived from adult CD34+ HSCs [156]. However, constitutive overexpression of KLF1 in hESCs resulted in reduced proliferation and differentiation into HSCs. To overcome this, they induced expression of KLF1 only when HSCs are present in their differentiation culture. They observed that the temporal expression of KLF1 in hESCs and hiPSCs improved erythroid lineage commitment and differentiation, and erythroblast enucleation. This study highlights the frequently neglected point that merely ensuring expression of erythropoiesis-related genes is inadequate in successfully generating iPSC-erythrocytes. Genetic approaches that constitutively express erythroid genes from the early stages of in vitro differentiation might be counteractive, and instead, precise temporal control of the up- and downregulation of erythroid genes should be employed where possible.

Other transcriptomic and proteomic comparisons between iPSC-erythroblasts and adult ones revealed differentially expressed genes/proteins that are involved in the autophagolysosomal pathway (VCPIP1, TRIM58), WNT/β-catenin pathway (GSK3α), cell-cycle regulation (PITX1) [152] and cytoskeleton remodeling (tubulin β-2A, CTNNA1, MAP1A/B and MARCKS) [157]. These pathways have been shown to play a role in definitive erythropoiesis [132], especially enucleation [158,159]; therefore, genetic interventions targeting these pathways should be explored in future studies. Another cytoskeletal protein of interest, vimentin, acts as a nucleus anchor in erythroblasts [160] and is lost during differentiation of mouse primitive and definitive erythroblasts, coinciding with enucleation [161,162]. Alternatively, avian erythrocytes, which do not undergo enucleation, retained vimentin expression throughout differentiation [163]. Erythroid cells differentiated from hiPSCs and hESCs still expressed vimentin up until the orthochromatic erythroblast stage, and in a small subset of those cells that did enucleate, vimentin was undetectable [164]. These data, although merely correlative, suggest that the loss of vimentin is essential for initiating enucleation and knocking-down vimentin expression during iPSC-erythroblast differentiation is a promising solution to the poor enucleation rate.

## 4. Generating Clinically Suitable iPSC-RBCs for Transfusion

### 4.1. GMP-Compliant RBC Products (Feeder and Serum-Free, Xenogeneic-Free)

As the molecular and genetic shortcomings of in vitro RBC products are slowly being solved, iPSC-derived therapies are approaching clinical translation. To date and to our knowledge, no clinical trial involving iPSC-derived RBCs has been attempted. Realizing the therapeutic potential of these cells will necessitate overcoming major obstacles of both a biological and engineering nature, such as the aforementioned problems of obtaining cells that more strictly resemble adult RBCs and exploring enucleation and hemoglobin switching. Additionally, from a process engineering perspective, strict adherence to good manufacturing practices (GMP) will need to be observed in order to ensure a robust and reproducible process with minimal chance of immunogenic effects when administered to patients. Research into RBCs has thus far relied extensively on xenogeneic/undefined components, such as bovine serum albumin (BSA) and feeder cells [62,69], which could also introduce variability between batches in a large-scale production setting [165]. Ideally, differentiation processes should exclude such components and use fully defined media formulations. In 2020, Tursky et al. performed a side-by-side comparison of four feeder and serum-free protocols for HSPC generation from hiPSCs and theorized that a monolayer multistep method showed the most robust generation of CD34+ progenitor cells at a cost of US$89 per 10^6^ CD34+ cells generated [166]. Fully defined, xenogeneic-free processes are an inescapable requirement for clinical translation, as the undefined animal components commonly utilized in processes may lead to immune reactions if inadvertently introduced into patients.

### 4.2. Scaling Up iPSC-RBC Generation

A bioreactor culture of mammalian cells has become a mainstay of the biopharmaceutical industry. Bioprocesses using CHO cells have been scaled to the hundreds of liters for the production of monoclonal antibodies (mAbs), bringing with them valuable lessons for the bioprocessing conditions that are most favorable to the fragile mammalian cells. This experience is now being applied to the emerging field of stem cell therapeutics, where the cells themselves are the desired product. However, before in vitro-produced RBCs can enter the public space as a viable therapeutic product, several key challenges must be addressed [91,167]. RBC differentiation processes (see Table 1) have advanced remarkably since their conceptualization, with many groups transitioning from 2D monolayer cultures to scalable 3D bioreactor settings. Below, we discuss the key limitations that they must now tackle on the road to a commercial RBC platform.

#### 4.2.1. In Vitro RBC Cost Evaluation

Although numerous advancements in the generation of RBCs from hiPSCs have been made in the last decade, bringing these processes to a commercial state would require substantial improvements in production methods. As previously mentioned, a single unit of blood contains 2 × 10^12^ enucleated red cells and is acquired by hospitals at a cost of approximately US$200–300 per unit [168]. However, initial uses of iPSC-derived RBCs would most likely compete more directly with phenotypically matched blood for patients with rare blood antigens, the price of which is estimated at approximately US$1000 [169]. Despite the acute shortage of healthy donors and the underlying assumption that critical life-saving material could be accepted at a premium, any in vitro RBC platform would have to be strongly competitive with these estimates, else they would be considered non-viable as business opportunities. Unfortunately, most in vitro RBC production processes conclude that the generation of one blood unit could carry a cost of US$15,000 or more, with some estimates reporting even 5 times higher [107,168,169].

High manufacturing costs for RBC products can be attributed to several key factors. Firstly, differentiation and expansion of hiPSCs for RBC production must transition from traditional 2D tissue culture flasks to more complex and controllable 3D vessels. Giarratana and colleagues performed studies demonstrating the ability of CB-isolated HSPCs to differentiate into fully mature and functional RBCs in a 2D setting, producing 1.4 × 10^10^ cells per flask (T-75) over the culture duration of 20 days. However, in practical terms, scale-up of this method would be unfeasible, as a single unit of blood would necessitate the use of ~150 flasks [38].

Another issue is that current published differentiation protocols overwhelmingly rely on commercial medium supplemented with complex cytokine cocktails, which in turn can vary significantly depending on the stage of differentiation. In order to transition through the various stages with high efficiency, media and cytokines are replenished often and kept at high concentrations. In particular, the availability of albumin and transferrin has been highlighted as a bottleneck due to the large quantities necessary for efficient cell maturation [75]. To complicate matters further, any bioprocess aimed at clinical trials must involve the use of chemically defined/xeno-free conditions in order to reduce the risk of introducing foreign materials into patients [170].

Ultimately, due to the high price of reagents, any cost-efficient process must be developed in a way that is able to achieve and maintain ultra-high densities of cells without negatively impacting their differentiation and maturation capacity, thus enabling maximum utilization of the supplied media. Traditional mammalian cell bioprocesses are considered high density at 10^7^ cells per mL [171], which for one unit of blood would translate into a 200 L culture volume at the maturation stage. This would be vastly cost-prohibitive even if the stages leading to maturation are discounted. Therefore, a successful process must be able to achieve ultra-high cell densities of 4–5 × 10^8^ cells/mL to generate sufficient cells for harvest in the smallest volume possible. If such a density could be achieved, one unit of blood could thus be produced in 4 L of medium, a 50-fold reduction.

#### 4.2.2. Bioprocess Intensification

Optimization of the iPSC-RBC platform should ideally tackle both the engineering and scientific aspects of the production bottleneck. Due to the relative immaturity of the field as compared to traditional mammalian bioprocessing, hiPSC cultivation has not yet been fully explored at the large industrial scale. Traditionally expanded as monolayers in tissue culture flasks, hiPSCs have proven challenging to expand to therapeutic numbers in suspension culture. Issues have arisen from low cell growth rates to shear stress concerns. In order to overcome these challenges, intensification studies must be performed to determine the limits of each step of in vitro RBC generation when cultured in a bioreactor format, with regards to nutrient utilization, oxygen demand and waste production.

A lot of recent research has focused specifically on pluripotent cell expansion in 3D systems, whose advantages over 2D cultures include reduced manpower and space requirements, easier handling, reduced batch variability and more options for in-process monitoring and control of key parameters. Kwok et al. demonstrated aggregate suspension culture in 1 L bioreactors, achieving a 10-fold expansion over 7 days of culturing [172], while Abecasis showed a 19-fold expansion in a smaller 200 mL DasGip bioreactor system [173]. A microcarrier suspension has also been shown as a viable option for iPSC maintenance and expansion: our group previously performed studies using vitronectin and laminin-coated microcarrier beads, achieving a 15-fold expansion over 7 days in culture [174]. An impressive new study by Pandey and colleagues used a single-use 3 L BioBlu vessel to expand hiPSCs in 10 serial passages, culminating in a 93-fold overall expansion while maintaining pluripotency markers [175]. Despite these successes, studies are ongoing to determine the optimal conditions for hiPSC cultivation using novel reactor configurations (such as the innovative vertical-wheel bioreactor), optimal stirring speeds, inoculation and harvesting strategies [176,177].

HSPC and erythroid culture have not yet been explored to such a degree. However, some notable studies have demonstrated a high-density culture of erythroid cells in suspension format. Bayley and colleagues performed promising studies in 2017 using umbilical cord CD34+ cells and reported that metabolic stress, mass and energy transfer and other conventional engineering constraints should theoretically allow for ultra-high densities of erythroblasts and enucleated red cells [171]. The study also demonstrated the very low oxygen consumption of erythroblasts in STRs with better maturation when agitated, although whether this is reproducible for iPSC-derived erythroblasts remains to be seen. Other groups have published methods with defined, GMP-compliant conditions for RBC expansion from PBMCs [51] and donor HSCs with high levels of enucleation and expansion [41]. Sivalingam et al. achieved cell densities of 2 × 10^12^ in suspension platforms by progressively scaling up the culture [72]. Here, we propose a practical solution to progressively expand hiPSCs and differentiate them into erythroid cells in increasing volumes of bioreactor cultures (see Figure 2).

Finally, innovative cultivation strategies, such as hollow fibers for continuous RBC harvesting, may also provide a viable option if process bottlenecks regarding expensive media formulations can be overcome. Such systems can provide a more in vivo-like niche for cell growth and interaction, which could be beneficial for cell maturation [178]. Their benefits include improved nutrient and gas exchange and relatively low shear stress. However, such systems have not been researched as extensively as traditional bioreactors and have not yet been explored for larger scale generation of blood cells.

Regardless of the production method employed, since the final product of the RBC differentiation processes are the cells themselves, mature enucleated cells must be separated efficiently from other constituents of the final harvest. Chief among these are extruded nuclei, otherwise known as the pyrenocytes, and cells that have failed to enucleate entirely. Large-scale cell separation systems must be employed in place of traditional leukocyte reduction filters to process the large amounts of media involved. To this end, microfluidic devices are being investigated for downstream processing abilities. In 2020, Guzniczak et al. demonstrated a two-stage separation process consisting first of a microfluidic chamber that employs deformability-based RBC sorting followed by a membrane filtration step for an overall cell purity of 99% in the final product [179]. In the same year, Zeming and colleagues investigated both Dean Flow Filtration (DFF) and Deterministic Lateral Displacement (DLD) sorting using a microfluidic approach and demonstrated that both methods were capable of achieving higher cell recovery over traditional fluorescence cell sorting [180]. Studies such as these are vital towards the development of clinically relevant bioprocesses, not just for blood production, but for many other systems with cellular products.

### 4.3. Expression of Fetal vs. Adult Hemoglobin

One final, yet potentially significant issue is the tendency of hiPSC-derived RBCs to express predominantly fetal and embryonic globins (HbF, HbE) instead of the expected adult ones (HbA). Since fetal hemoglobin binds oxygen more tightly than its adult counterpart, this results in a significant left shift on an oxygen dissociation curve analysis compared to adult RBCs [72]. Research into the control of hemoglobin switching is ongoing and key findings have already begun to surface. For instance, Yang and colleagues measured lower expression of KLF1 in hESC-derived erythroblasts as compared to ones derived from adult-isolated CD34+ cells. The group demonstrated that induction of KLF1 in differentiated pluripotent cells can help to enhance final cell maturity [156]. KLF1 itself is known as a DNA-binding protein that is involved in the regulation of cell identity in erythroid cells, including roles in heme and globin synthesis, where it is a critical factor in the fetal-to-adult globin switch [181]; KLF1 haploinsufficiency has been implicated in hereditary fetal hemoglobin persistence in adults [182]. However, it has been demonstrated that HbF-expressing iPSC-RBCs are capable of performing a switch to HbA expression in vivo when injected into NOD/SCID mice [68]. Furthermore, it has been noted that HbF-related anemia is not found in patients with hereditary persistence of fetal hemoglobin, thus indicating that such cells could still possess significant therapeutic value [183]. Additionally, the expression of HbF in iPSC-derived RBCs may find a niche application in the transfusion of pre-term babies as fetal blood rather than adult blood is needed at this stage of development.

## 5. Novel Applications for hiPSCs-Derived, Engineered RBCs

Amidst the research of large-scale production of hiPSCs-derived, engineered RBCs, and the shift in focus to optimizing enucleation, a viable bioprocess for blood transfusions shows great promise. However, it is also important to think about their other potential scientific applications.

### 5.1. Genetic Amenability and Application of hiPSC-Derived RBCs for Basic Research

The carbohydrates and proteins on the RBC surface are responsible for different blood groups, which give a unique RBC profile to the individual. A total of 43 blood group systems have been identified and recognized by the International Society for Blood Transfusion (ISBT). Presence of diverse blood groups aggravate the challenges in supporting blood demand worldwide, as blood group polymorphisms on RBCs might lead to the development of rare blood groups or blood disorders, and a switch in RBC susceptibility to parasitic infection. Due to the scarce blood supply for transfusion and infection studies, iPSC-derived RBCs have seen an upward trend for erythrocytic research. hiPSCs are immortal and have clonal growth, making them highly suitable for genetic editing. Hence, RBCs with specific blood group antigens can be easily generated via genetic modification of hiPSCs.

Prior to blood transfusion, crossmatching is performed for donor and patient blood to determine their compatibility in order to avoid undesirable transfusion reactions, such as hemolysis of transfused blood. Transfusion for genetic blood disorders or rare blood groups is challenging due to limited donor availability and storage limitations. hiPSCs can be a boundless supply of autologous RBCs for such transfusions by reprogramming a patient’s somatic cells into erythrocytes. It is important to note that, while iPSC-RBCs will likely suffer from similar storage limitations as donor blood, successful bioprocesses will be able to augment shortages in a timely manner with RBCs of specific blood types, regardless of donor availability. Park et al. has successfully generated autologous iPSC-derived RBCs for rare blood type from peripheral blood mononuclear cells without any chromosomal mutations [184]. As O-negative blood is the universal blood type for transfusion, another group of researchers has also established hiPSCs from primary bone marrow CD34+ cells of an O-negative blood donor [185]. However, more basic research is needed to overcome all the hurdles in generating iPSC-derived RBCs before they are feasible for blood transfusion.

On the other hand, RBCs can be infected by erythrocytic parasites, such as *Plasmodium*, *Babesia*, *Bartonella* and *Toxoplasma* species [186]. In the context of malaria, the invasion of *Plasmodium* species is dependent on the presence of certain blood group antigens, such as the MNS (CD235a and CD235b), Gerbich (CD236), Knops (CD35), Ok (CD147) and Duffy (CD234) blood group systems [187,188]. However, blood group antigens responsible for erythrocytic invasion of most of the *Plasmodium* species are still not known [189]. There is also limited information on the blood group antigens for invasion of *Babesia* [190], *Bartonella* [191] and *Toxoplasma* species [192]. Consequently, the manipulation of blood group antigens on RBCs via hiPSC genetic modification allows studies of the respective blood group antigens’ role in parasite invasion mechanisms.

SARS-CoV-2 virus has been shown to infect RBC indirectly. The oxygen-saturation level of COVID patients with severe symptoms tends to be lower [193]. This is on account of the virus infection disrupting the RBC membrane structure, which in turn reduces the RBC efficiency in off-loading oxygen [194]. The genetic amenability of iPSC-derived RBCs would allow easy manipulation of essential proteins involved in the membrane structure, thereby facilitating infection mechanism studies and the drug development process.

### 5.2. Universal hiPSCs-Derived RBC-EVs for Medical Treatments (Drug Delivery Vehicles and oncomiR Gene Editing)

During erythroid expansion, RBCs release both exosomes and plasma membrane derived EVs (ectosomes) [195]. RBC exosomes are formed only during the development of RBCs in bone marrow and are released following the fusion of microvesicular bodies with the plasma membrane. On the other hand, RBC EVs are generated during normal aging of RBCs in circulation by budding of the plasma membrane due to complement-mediated calcium influx, followed by vesicle shedding.

In 2018, Usman et al. was able to demonstrate efficient delivery of both long and short RNAs into cells with RBC EVs as compared to two commonly used transfection agents, with full functionality and stability in their liquid and solid cancer ex vivo and in vitro assays [196]. They were also able to target a specific oncomiR gene, not only via steric blocking, but also via CRISPR–Cas9 genome editing, which had hitherto not been shown. This research revealed the immense potential for non-toxic and non-immunogenic drug delivery in cancer research with RBC EVs via matching of blood types unlike most synthetic transfection reagents counterparts, such as adenoviruses, adeno-associated viruses, lentiviruses and nanoparticles. Additionally, the integrity and efficacy of the produced RBC EVs remains unaffected over multiple freeze–thaw cycles, showing their potential for future pharmaceutical product development [196].

Therefore, this finding expands the array of benefits that could be derived from large-scale erythroid bioprocess. Firstly, the culture media, which is generally considered bio-industrial waste, could be purified for its RBC EV content as a downstream clinical product. Secondly, universal hiPSCs-derived RBC-EVs obtained from a large-scale bioprocess serve as a better source due to its immense quantity generated per bioreactor run.

## 6. Conclusions

Recent advancements in the past decade for in vitro generation of RBCs seem to suggest great promise for efficient high-density erythroid bioprocesses despite the challenges faced. The applications for RBCs from hiPSCs make them very attractive options to pursue as eventual substitutes to donor blood.

While the enucleation bottleneck still remains largely unsolved, we have begun to gain track in understanding its mechanisms and found several innovative ways to mimic enucleation microenvironments through co-culture or genetic approaches (activation or inhibition of transcription factors). Efficient enucleation on a 3D platform has also been established, which allows for potential scale up. Moving forward, investigations on the macroenvironmental conditions, such as agitation, dissolved oxygen levels, supplementation of protective polymers and small molecules during enucleation, could reveal beneficial results for a more industrially compatible bioprocess.

## Figures and Tables

**Figure 1 ijms-22-09808-f001:**
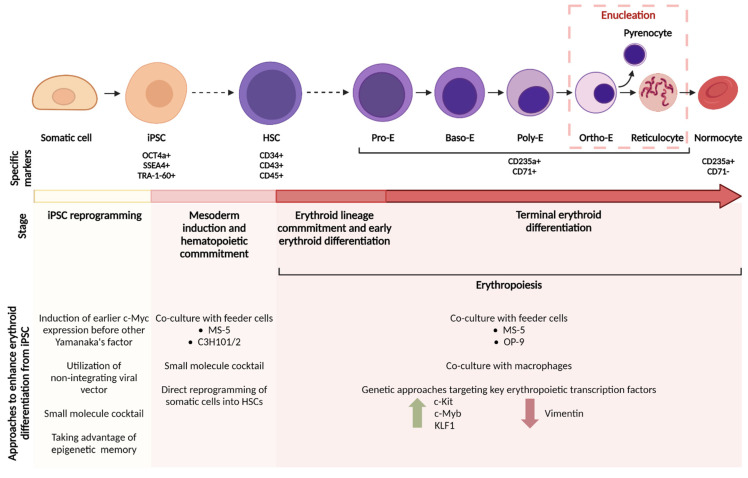
Approaches to enhance erythroid differentiation from iPSC. The production of iPSC-derived erythrocytes involves multiple differentiation steps, from mesoderm induction, hematopoietic commitment, erythroid lineage commitment to erythroid differentiation, each identified by specific sets of markers. Different approaches involving genetic modifications or external factors are needed to improve erythropoiesis at each stage. iPSC, induced pluripotent stem cell; HSC, hematopoietic stem cell; Pro-E, proerythroblast; Baso-E, basophilic erythroblast; Poly-E, polychromatic erythroblast; Ortho-E, orthochromatic erythroblast. Created with BioRender.

**Figure 2 ijms-22-09808-f002:**
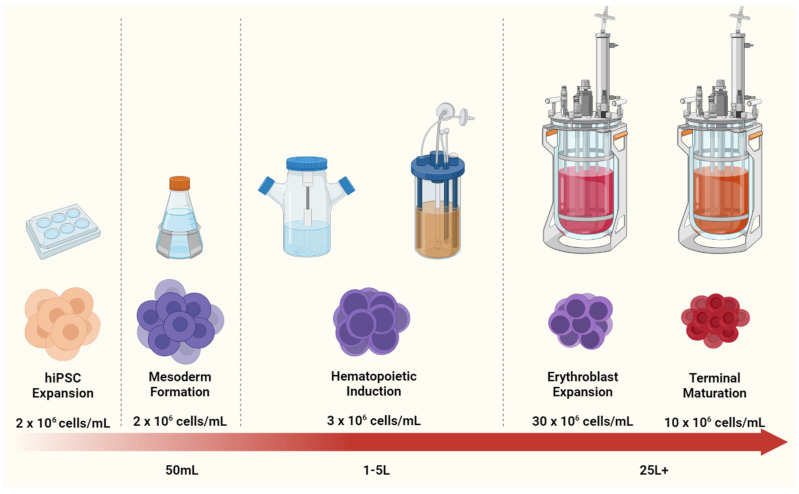
Scale-up of iPSC-RBC production in a suspension culture. Adapting each stage of the hematopoietic and erythroid specification in the suspension culture provides a starting point to achieve larger volumes and densities. A single 6-well ULA plate (left section) can provide enough cells for expansion and downstream differentiation in increasing volumes, from shake flasks to larger vessels, such as spinner flasks and lab-scale bioreactors (middle section), and eventually transitioning into large, controlled, stirred bioreactors (right section) for high-density erythroid culture and maturation of the cell product. Created with BioRender.

## Data Availability

All data are available in the review.

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
