# Peer review of "Industrially Compatible Transfusable iPSC-Derived RBCs: Progress, Challenges and Prospective Solutions"

_ijms, 2021, doi:10.3390/ijms22189808_

Round 1

Reviewer 1 Report

In this review, Lim and coll. describe RBC generation from iPSCs and discuss current advances and challenges. Overall, the manuscript is well designed and covered most of the important literature.

Remarks:

  • Revise the following sentences for better clarifications; Lines 154-157, 192-193, 318-320, and 327-328.
  • Line 162: Define “rapidly” by giving an exact timeframe.
  • Line 217: Although authors define hematopoietic development later in the review, as they mention definitive erythrocytes in this sentence, it is better to describe definitive and primitive erythrocytes and their differences here.
  • Some critical articles are missing in different parts. Consider adding them

PSC survival (PMID: 33941937), HCS from PSCs (PMID: 28514439 and PMID: 23670574).

  • To provide consistency, after defining the abbreviations use always the abbreviations (i.e iPSCs but not induced pluripotent stem cells after defining the iPSCs)
  • Line 317: To make the titles similar, make it Non-integrating methods.
  • Line 411: Although the statement is correct, it is not the definition for HSCs with engraftment capacity.
  • There are some grammatical mistakes. The manuscript needs to be checked for its language (i.e Line 226)

Reviewer 2 Report

Lim and colleague reviewed the protocol for RBC induction form ESC and iPSCs. The article contains current assays for RBC production and explained from hematopoietic development through industrial RBC culture with many references and information. I suggest few points to improve the article.

  1. It should be appreciated if author add the figure for the general step of RBC inducing from iPSC (FACS plot e.g.).

  1. Using iPSCs of one advantage is that for alleviation of immune rejection when the tissue transplants to. Is there some report for such a evidence at iPSCs produced RBCs compared with hESCs/hCBs produced RBCs (mouse is fine)?

  1. LINE 366, The author concluded BM origin of iPSC is easy to reproduce hematopoietic cell including RBCs. However, in anemia situation, it’s usually observed decreasing hematopoietic cells due to bone marrow failure. Is there some report of evidence for more easy to reprograming lineage cell to hematopoietic cell? 

  1. LINE 406, Should be added the explanation of primitive and definitive differences (hematopoiesis organ e.g.).

Reviewer 3 Report

In this paper is summarized the current problems of blood transfusion practice, requirement of blood substitutes, the trials of producing artificial and biological RBCs, and the recent successful achievement of iPSC-derived RBCs and the remained hurdles for industrialization. I think this topic is very important for the future human health and welfare, and should be attractive for the readers of Int. J. Mol. Sci. However, some revisions would be required as shown below.

1) page 2, The authors summarize the storage-induced lesions in donor blood. However, the authors should admit that the same problems will be anticipated for the biological iPSC-derived RBCs in the future. They may also have short shelf fives and cannot overcome fragile properties of RBCs.

2) page 3, line 115-117. “Despite the span of many decades, successful attempts in generating blood substitutes were futile due to the toxic attributes and low retention rates in circulation.”

In the section 1.3., the authors summarize the history of artificial oxygen carriers made of PFC and Hb (Hb based oxygen carriers, HBOCs), focusing on several unsuccessful clinical trials, and they conclude they were “futile”. However, as far as I know high research activities continue in the research field of PFC and HBOCs worldwide to overcome the drawbacks (see the recent references below should be included). I think such continuing efforts should be included in this section, showing the respectfulness to all the researchers who aim at realization of blood substitutes solving the common problems. Please note that blood substitutes include not only iPSC-derived RBCs, but also PFC and HBOCs. The authors should admit that the development of iPSC-derived RBCs has not reached to the stage of a major clinical trial yet.

(Refs)

Krafft MP, Riess JG. Therapeutic oxygen delivery by perfluorocarbon-based colloids. Adv Colloid Interface Sci. 2021 Aug;294:102407.

Sakai H, et al. Translational research of hemoglobin vesicles as a transfusion alternative. Curr Med Chem. 2021 Apr 12. doi: 10.2174/0929867328666210412130035.

Chang T, et al. Eds. “Nanobiotherapeutic Based Blood Substitutes”, World Scientific Pte Ltd., Singapore 2021.

Abu Jawdeh BG, et al. A phase Ib, open-label, single arm study to assess the safety, pharmacokinetics, and impact on humoral sensitization of SANGUINATE infusion in patients with end-stage renal disease. Clin Transplant. 2018 Jan;32(1).

and more......

3) Page 5, line 196- The authors summarized the hESCs and iPSC. The next related important papers should be included.

(ref)

Ma F et al. Generation of functional erythrocytes from human embryonic stem cell-derived definitive hematopoiesis. Proc Natl Acad Sci U S A. 2008 Sep 2;105(35):13087-92.

Niwa A, et al. A novel serum-free monolayer culture for orderly hematopoietic differentiation of human pluripotent cells via mesodermal progenitors. PLoS One. 2011;6(7):e22261.

4) page 14, Generating Clinically Suitable iPSC-RBCs for Transfusion.

If there is any attempt of clinical trial in this research field, it should be included in this section before discussing about cost effectiveness, and also in the abstract, which will be very informative for the readers. If there is no attempt of clinical trial, please describe clearly the hurdles for the development.

5) Figure 2 shows the reactors up to 25L scale. Is this scale enough for industrialization?

6) iPS-derived platelet is successfully developed. Please consider to include it somewhere as one of the blood substitutes.

Ito Y, et al. Turbulence Activates Platelet Biogenesis to Enable Clinical Scale Ex Vivo Production. Cell. 2018 Jul 26;174(3):636-648.e18.

Round 2

Reviewer 3 Report

The manuscript is revised well according to the comments from the reviewer. 

One minor point;

In Table 1, the third column, the words are broken such as

Monolaye -  r,  Supensio -  n

In the fourth column,

KLF1-activat -  ed

Probably they should be hyphenated, or the font size should be reduced.